# Penile-Sparing Surgery for Tumour Recurrence after Previous Glansectomy/Partial Penectomy: Treatment Feasibility and Oncological Outcomes

**DOI:** 10.3390/cancers15194807

**Published:** 2023-09-29

**Authors:** Gennaro Musi, Filippo Molinari, Francesco A. Mistretta, Mattia Luca Piccinelli, Sonia Guzzo, Marco Tozzi, Elena Lievore, Oskar Blezien, Matteo Fontana, Antonio Cioffi, Daniela Cullurà, Elena Verri, Maria Cossu Rocca, Franco Nolè, Matteo Ferro, Ottavio de Cobelli, Stefano Luzzago

**Affiliations:** 1Department of Urology, IEO European Institute of Oncology, IRCCS, Via Ripamonti 435, 20141 Milan, Italy; gennaro.musi@ieo.it (G.M.); filippo.molinari@ieo.it (F.M.); francescoalessandro.mistretta@ieo.it (F.A.M.); mattialuca.piccinelli@ieo.it (M.L.P.); sonia.guzzo@ieo.it (S.G.); marco.tozzi@ieo.it (M.T.); elena.lievore@ieo.it (E.L.); oskar.blezien@ieo.it (O.B.); matteo.fontana@ieo.it (M.F.); antonio.cioffi@ieo.it (A.C.); matteo.ferro@ieo.it (M.F.); ottavio.decobelli@ieo.it (O.d.C.); 2Department of Oncology and Hematology-Oncology, Università degli Studi di Milano, 20122 Milan, Italy; 3Department of Medical Oncology, Division of Urogenital and Head and Neck Tumours, IEO European Institute of Oncology, IRCCS, Via Ripamonti 435, 20141 Milan, Italy; daniela.cullura@ieo.it (D.C.); elena.verri@ieo.it (E.V.); maria.cossurocca@ieo.it (M.C.R.); franco.nole@ieo.it (F.N.)

**Keywords:** glansectomy, partial penectomy, laser ablation, excision, penile cancer

## Abstract

**Simple Summary:**

To date, no specific analyses focusing on penile-sparing surgery for local tumour recurrence after previous glansectomy or partial penectomy have been reported. We addressed this void and we considered a retrospective series of consecutive patients treated at a single institution. We focused on: (1) treatment feasibility, (2) complications, and (3) oncological outcomes.

**Abstract:**

We tested the feasibility and oncological outcomes after penile-sparing surgery (PSS) for local recurrent penile cancer after a previous glansectomy/partial penectomy. We retrospectively analysed 13 patients (1997–2022) with local recurrence of penile cancer after a previous glansectomy or partial penectomy. All patients underwent PSS: circumcision, excision, or laser ablation. First, technical feasibility, treatment setting, and complications (Clavien–Dindo) were recorded. Second, Kaplan–Meier plots depicted overall and local recurrences over time. Overall, 11 (84.5%) vs. 2 (15.5%) patients were previously treated with glansectomy vs. partial penectomy. The median (IQR) time to disease recurrence was 56 (13–88) months. Six (46%) vs. two (15.5%) vs. five (38.5%) patients were treated with, respectively, local excision vs. local excision + circumcision vs. laser ablation. All procedures, except one, were performed in an outpatient setting. Only one Clavien–Dindo 2 complication was recorded. The median follow-up time was 41 months. Overall, three (23%) vs. four (30.5%) patients experienced local vs. overall recurrence, respectively. All local recurrences were safely treated with salvage surgery. In conclusion, we reported the results of a preliminary analysis testing safety, feasibility, and early oncological outcomes of PSS procedures for patients with local recurrence after previous glansectomy or partial penectomy. Stronger oncological outcomes should be tested in other series to optimise patient selection.

## 1. Introduction

Penile-sparing surgery (PSS) is the recommended strategy for patients with localized penile cancer, whenever feasible, due to its efficacy to remove the entire tumour while preserving as much of the penis as possible [1]. PSS is associated with higher rates of local recurrence (10–55%), but similar overall survival, compared with partial or radical penectomy [2,3,4,5]. Several PSS procedures have been recently developed for penile cancer patients, varying from less invasive techniques, such as topical chemotherapy or laser ablation, to more aggressive treatments like glansectomy or partial penectomy [6]. However, despite the rates of local recurrence varying according to the PSS technique used [5,7], even with glansectomy approximately 4–12.8% of patients experience local recurrence during follow-up [8,9]. Historically, total amputation has been offered to those patients who exhibit local recurrence after previous glansectomy/partial penectomy, compromising the functional results of previous PSS [10]. However, some of those patients with localised recurrence could be amenable to repeat PSS procedures, without compromising oncological control of the disease. This said, to the best of our knowledge, no specific analyses focusing on this management strategy exist to date and only sporadic cases have been reported by previous authors [11,12].

We hypothesised that a group of selected patients with disease recurrence after previous glansectomy/partial penectomy could be safely treated with a new PSS procedure.

To address this void, we focused on a consecutive series (1997–2022) of patients with penile cancer recurrence after glansectomy or partial penectomy and we tested the surgical feasibility of another PSS and subsequent recurrence rates over time.

## 2. Materials and Methods

### 2.1. Patients 

This study respected the ethical guidelines of the Declaration of Helsinki. A retrospective analysis of all penile cancer patients treated at our centre between 1997 and 2022 (n = 263) was performed and we selected men submitted to glansectomy or partial penectomy (n = 174; 66%). Of those, we focused on patients who exhibited local recurrence during follow-up (n = 35; 20.1%) and who were treated with total penectomy (n = 22; 62.9%; Appendix A) or PSS (n = 13; 37.1%; Table 1). The latter were included in the final analyses.

### 2.2. Penile-Sparing Surgery

During the study period, several PSS techniques were used, based on the location and the dimension of the lesion, the preference of the surgeons, and the availability of the technologies. PSS consisted of any of the following: circumcision, local excision [12], or laser ablation (either CO2 [13] or thulium–yttrium–aluminium–garnet (Tm:YAG) lasers [14]).

Follow-up after PSS respected the European Association of Urology (EAU) guidelines [1]. Physical examination was performed every 3 months in the first 2 years and every 6 months in the following 3 years. Patients were also advised to perform regular self-examination. Follow-up imaging scans also respected the EAU guidelines [1].

### 2.3. Variable Definitions and Statistical Analyses

Variables recorded included: age at surgery, year of diagnosis, Charlson Comorbidity Index (CCI), Body Mass Index (BMI), smoking status, HIV and HPV infections, type of previous surgery, tumour size (mm), lesion site, type of surgery, margin status, TNM stage, and tumour grade. Surgical complications were graded according to the Clavien–Dindo classification [15]. Descriptive statistics relied on tests of medians and proportions for, respectively, continuously coded and categorical variables. We conducted a two-step analysis. 

First, we focused on the technical feasibility of PSS after glansectomy or partial penectomy. Specifically, we registered the PSS technique used, as well as the treatment setting (outpatient vs. inpatient) and complications. Second, we tested for overall disease recurrences (either distant vs. regional vs. local) as well as local recurrences over time. Here, Kaplan–Meier plots were used. All statistical tests were two-sided with a level of significance set at *p* < 0.05 and were performed using the R software environment for statistical computing and graphics (version 3.4.1; http://www.r-project.org/).

## 3. Results

### 3.1. Descriptive Analyses (Table 1)

The median (interquartile range: IQR) age at surgery was 60 (53–63) years (Table 1). CCI was 1, 2, and ≥3 in, respectively, in four (30.5%), four (30.5%), and five (39%) patients. The median (IQR) tumour size at the time of the previous surgery was 25 (20–30) mm. In consequence, 11 (84.5%) vs. 2 (15.5%) patients underwent glansectomy vs. partial penectomy. Histology at initial surgery was the following: squamous cell (84%) vs. verrucous (7.5%) vs. epidermoid (7.5%) carcinoma. Moreover, T-stage stratification revealed the following distribution: Tx (7.5%) vs. Tis (7.5%) vs. T1 (39%) vs. T2 (46%). Additionally, 7.5% vs. 15% vs. 30.5% vs. 47% of men had Gx vs. G1 vs. G2 vs. G3 tumour grade, respectively. Last, only one patient (7.5%) had previous N1 disease.

### 3.2. Perioperative Findings (Table 2) 

The median (IQR) time from glansectomy/partial penectomy to disease recurrence was 56 (13–88) months. The median (IQR) tumour size was 7 (5–15) mm. Overall, 10 (77%) vs. 2 (15.5%) vs. 1 (7.5%) recurrences were located at, respectively, neoglans vs. neoglans + foreskin vs. distal urethra. Two exemplificative cases are depicted in Figure 1. In consequence, six (46%) vs. two (15.5%) vs. five (38.5%) patients were treated with, respectively, local excision vs. local excision + circumcision vs. laser ablation. PSS procedures were performed in outpatient vs. inpatient settings in 12 (92.5%) vs. 1 (7.5%) cases. Specifically, patient 13 had an 18 mm recurrence at the level of the neoglans and was treated with wide local excision under general anaesthesia. This patient had a length of stay of 4 days and required antibiotic therapy for a Clavien–Dindo grade 2 complication. All other patients did not experience complications after PSS. Final histology was available for 10 (77%) men. All tumours were squamous cell carcinoma. Four (30.5%) vs. one (7.5%) vs. one (7.5%) vs. four (30.5%) tumours were PeIN vs. Ta vs. Tis vs. T1, respectively. Last, five (38.5%) vs. four (30.5%) vs. one (7.5%) vs. three (23.5%) lesions were Gx vs. G1 vs. G2 vs. G3, respectively.

### 3.3. Findings at Follow-Up (Table 3)

The median (IQR) follow-up time was 41 (10–72) months. During follow-up, three (23%) vs. four (30.5%) patients experienced local (Figure 2a) vs. overall (Figure 2b) recurrence, respectively. Specifically, patient 3 was only treated with bilateral inguinal lymph node dissection for isolated nodal recurrent disease 3 months after PSS. Of all patients that exhibited local recurrence (n = 3), two (66.5%) vs. one (33.5%) underwent penectomy vs. wide local excision. Specifically, patient 9 experienced a pT1G2 squamous tumour 31 months after PSS and required penectomy and concomitant sentinel lymph node dissection. Moreover, patient 10 was treated with penectomy for a pT2G3 verrucous carcinoma that recurred 10 months after laser ablation. Last, patient 2 experienced a pT1aG1 squamous tumour that recurred 7 months after PSS. Wide local excision and sentinel lymph node dissection were performed. No patients died during the study period. Local and overall recurrence survival rates for patients previously treated with glansectomy/partial penectomy who underwent another PSS vs. radical penectomy for disease recurrence are depicted in the Appendix A.

## 4. Discussion

Glansectomy and partial penectomy are effective treatments for localised penile cancer, permitting oncological control over time, while simultaneously preserving patient sexual and urinary functions [16]. Unfortunately, approximately 4–12.8% of patients treated with these treatment modalities experience local recurrence during follow-up [8,9]. Historically, radical penectomy has been considered the treatment of choice for recurrent disease in these cases. However, a subgroup of patients with limited local recurrence could be considered for a new PSS procedure, without compromising the functional outcomes of previous conservative surgery [10]. To date, no systematic analyses have been conducted and only sporadic cases were previously reported by some authors [3,4]. We analysed 13 consecutive patients treated with PSS for local recurrence after glansectomy or partial penectomy between 1997 and 2022 with a specific focus on (1) technical feasibility and (2) oncological outcomes. Our results show several important findings. 

First, of all patients who experienced local recurrence after glansectomy or partial penectomy, 37% were treated with PSS. This percentage appears to be encouraging since approximately a third of patients could avoid immediate penile amputation. Moreover, this percentage could also be underestimated since the gold standard treatment for local recurrence after glansectomy/partial penectomy is represented by total penectomy. Due to the lack of specific recommendations for PSS after glansectomy/partial penectomy, the accurate selection of candidates appears to be a key factor. Unfortunately, due to the lack of information about postoperative surgical margins and the small number of patients analysed, only hypothetical considerations could be derived from this analysis. Specifically, patient age, education, comorbidities, sexual life, and compliance with strict follow-up schemes appear to be crucial. Moreover, other tumour characteristics, such as a long time to recurrence from previous surgery, small lesion size, low tumour T stage and grade, as well as recurrence location should be considered. Indeed, in our series, compared with patients immediately treated with radical penectomy, patients treated with PSS had smaller and more superficial tumours. Moreover, the time to disease recurrence was significantly lower for patients submitted to total amputation. Last, surgeon experience and hospital volume appear to be important when recommending PSS for recurrent disease. However, other reports testing the oncological safety and technical feasibility of PSS after glansectomy/partial penectomy are urgently required to optimise patient selection and promote wider use of PSS for this patient category. 

Second, we demonstrated that PSS for local recurrence after glansectomy/partial penectomy is technically feasible. According to tumour characteristics, clinician preference, and availability of technologies, several PSS procedures could be safely performed in this patient category. Specifically, all our patients were treated with either excision [12] or laser ablation [13,14]. Moreover, the vast majority of the surgeries were performed in outpatient settings and only one patient with an 18 mm lesion necessitated general anaesthesia and hospital recovery. Additionally, only one Clavien–Dindo 2 complication that was easily treated with antibiotic therapy was observed. Our findings encourage the use of PSS for selected patients with disease recurrence after previous glansectomy/partial penectomy. However, future analyses should focus on other important outcomes such as operative time, patient satisfaction, and sexual and urinary function before recommending implementation in daily practice. Moreover, until clear evidence of the superiority of one PSS technique over the others is demonstrated, all PSS procedures should be encouraged for treating local recurrences after glansectomy or partial penectomy.

Third, we tested early oncological outcomes after PSS for local recurrence after glansectomy/partial penectomy. Here, we observed that 23% and 30.5% of patients experienced local and overall recurrence over time, respectively. Our findings could indicate the safety of PSS in local control of recurrent disease for this patient category. Specifically, all patients who experienced local recurrence after PSS (n = 3) were safely re-treated with penectomy or wide excision. Conversely, only one patient presented isolated nodal recurrence (N1) and necessitated bilateral inguinal lymph node dissection. Our results also indicate that immediate penile amputation, at the time of the first local recurrence after glansectomy/partial penectomy, could probably be avoided in this patient category and only offered to those who exhibit another local recurrence during time (salvage setting). However, we advocate for accurate patient selection and strict follow-up of patients who are candidates for these treatment modalities. Moreover, our findings should be considered exploratory at best, since only a limited number of patients (n = 13) over a long time span (1997–2022) were treated at one referral centre. Last, the limited follow-up available for this patient cohort (median: 41 months) is not enough for testing major oncological endpoints such as tumour progression and cancer-specific mortality. In consequence, we advocate testing the oncological safety of another PSS procedure for tumour recurrence after a previous glansectomy or partial penectomy in a series of patients with longer follow-up data.

Taken together, we reported the technical feasibility and oncological outcomes of PSS for local recurrence of patients previously treated with glansectomy or partial penectomy for penile cancer. We observed that approximately a third of patients with local recurrence could be treated with PSS without compromising oncological control of the disease. Moreover, for those patients who experience another recurrence over time, salvage penectomy could be safely offered.

Despite its novelty, our study has limitations. First, the current data are retrospective and influenced by inherent selection bias. Second, as previously stated, we were unable to fit multivariable Cox models predicting recurrence rates over time due to a low number of patients and events. Third, we created heterogeneity among patients by including several PSS techniques (real-life scenarios). Fourth, information about surgical margin status was unavailable after PSS [17,18,19]. Fifth, some important pathological features, such as lymphovascular invasion and T1 sub-classification (T1a vs. T1b) [20,21,22] were unavailable. Sixth, as previously stated, information about patient satisfaction and sexual and urinary function after PSS were not recorded. Last, we did not perform a systematic comparison between patients who were immediately treated with total penectomy at first local recurrence after glansectomy or partial penectomy vs. those re-treated with PSS. Specifically, we only reported Kaplan–Meier plots depicting local and overall recurrence survival rates in these two groups (Appendix A), without extensively discussing our misleading findings (i.e., lower overall recurrence rates in PSS-treated patients), which are, in our opinion, a product of selection bias.

## 5. Conclusions

We reported the results of a preliminary analysis testing safety, feasibility, and early oncological outcomes of PSS procedures for patients with local recurrence after previous glansectomy or partial penectomy. Stronger oncological outcomes should be tested in other studies to optimise patient selection.

## Figures and Tables

**Figure 1 cancers-15-04807-f001:**
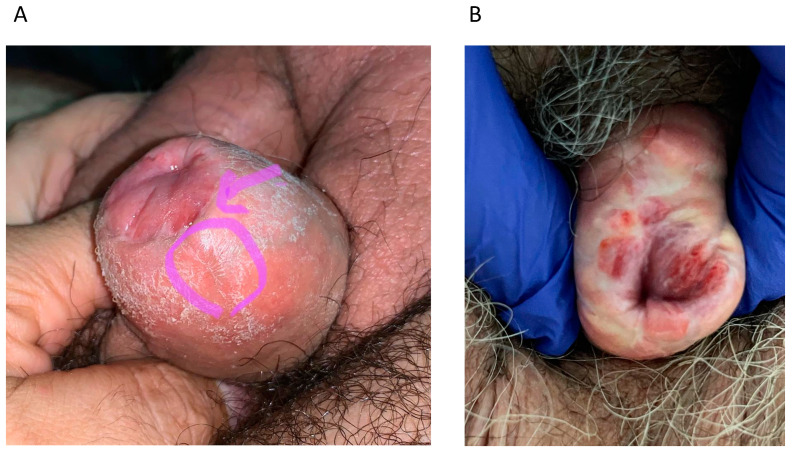
(**A**) Patient 1: Local recurrence at the level of the neoglans + foreskin at 56 months after glansectomy that underwent circumcision + excision. (**B**) Patient 3: Local recurrence at the level of the neoglans + foreskin at 9 months after glansectomy that underwent circumcision + excision.

**Figure 2 cancers-15-04807-f002:**
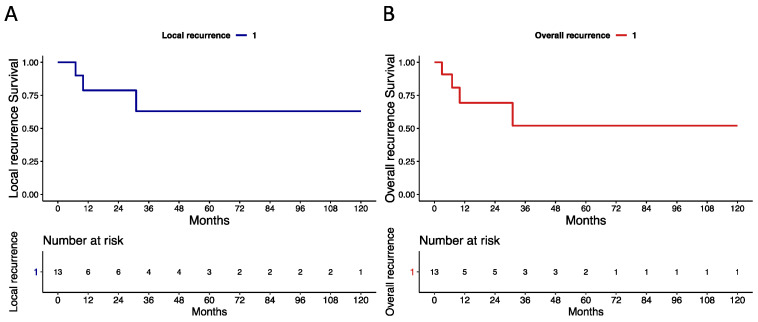
Kaplan–Meier plots depicting recurrence-free survival rates in 13 patients with recurrent penile cancer treated with penile-sparing surgery after previous glansectomy or partial penectomy (1997–2022). (**A**) Local recurrence. (**B**) Overall recurrence (any local, regional, or systemic recurrence).

**Table 1 cancers-15-04807-t001:** Clinical characteristics of 13 penile cancer patients, previously treated with glansectomy or partial penectomy, and subsequently treated with penile-sparing surgery for penile cancer recurrence between 2001 and 2022. Data are shown as medians for continuous variables or as counts and percentages (%) for categorical variables. IQR: interquartile range; CCI: Charlson Comorbidity Index; BMI: Body Mass Index; HIV: human immunodeficiency virus; HPV: human papillomavirus.

Age (Years)	Median (IQR)	60 (53–63)
**CCI, n (%)**	1	4 (30.5)
	2	4 (30.5)
	≥3	5 (39.0)
**BMI**	Median (IQR)	27 (26–32)
**Smoking**	No	8 (61.0)
	Yes	5 (39.0)
**Diabetes Mellitus**	No	10 (77.0)
	Yes	3 (23.0)
**HIV**	No/Unknown	12 (92.5)
	Yes	1 (7.5)
**HPV**	No/Unknown	12 (92.5)
	Yes	1 (7.5)
**Previous surgery, n (%)**	Partial/total glansectomy	11 (84.5)
	Partial penectomy	2 (15.5)
**Previous tumour size (mm)**	Median (IQR)	25 (20–30)
**Previous histology, n (%)**	Squamous cell	11 (85.0)
	Verrucous	1 (7.5)
	Epidermoid	1 (7.5)
**Previous T stage, n (%)**	Unknown	1 (7.5)
	Tis	1 (7.5)
	T1	5 (39.0)
	T2	6 (46.0)
**Previous tumour grade, n (%)**	Gx	1 (7.5)
	G1	2 (15.0)
	G2	4 (30.5)
	G3	6 (47.0)
**Clear margin (mm)**	Median (IQR)	4.5 (3–6)
**Previous N stage, n (%)**	Nx	4 (31.5)
	N0	8 (61.0)
	N1	1 (7.5)

**Table 2 cancers-15-04807-t002:** Perioperative findings of 13 penile cancer patients previously treated with glansectomy or partial penectomy, and subsequently treated with penile-sparing surgery for penile cancer recurrence between 2001 and 2022. iLND: inguinal lymph node dissection; sLND: sentinel lymph node dissection; N.A.: not available; PeIN: penile intraepithelial neoplasia.

Patient	Previous Surgery	Previous Histology	Age	Time to Surgery (Months)	Type of Surgery	Setting (LOS)	Lesion Site	Lesion Size (mm)	Histology	CLAVIEN– DINDO
1	Glansectomy + iLND	pT1N0G1 squamous	52	56	Circumcision + excision	Outpatient (1)	Neoglans + foreskin	N.A.	N.A.	0
2	Glansectomy + iLND	pT1N0G2 squamous	50	5	Excision	Outpatient (1)	Neoglans	N.A.	pTaNxG1 squamous	0
3	Glansectomy	pT2NxG2 squamous	53	9	Circumcision + excision	Outpatient (1)	Neoglans + foreskin	N.A.	N.A.	0
4	Partial penectomy	pTisNxG3 squamous	54	142	Excision	Outpatient (1)	Neoglans	16	pT1NxGx squamous	0
5	Glansectomy + iLND	pTxN0Gx squamous	75	13	Laser ablation	Outpatient (1)	Neoglans	6	pT1NxGx squamous	0
6	Glansectomy + iLND	pT1N0G3 squamous	34	151	Laser ablation	Outpatient (1)	Neoglans	5	N.A.	0
7	Partial penectomy + iLND	pT2N1G3 epidermoid	59	80	Excision	Outpatient (1)	Neoglans	7	PeINNxG1 squamous	0
8	Partial glansectomy + iLND	pT2N0G3 squamous	60	91	Excision	Outpatient (1)	Urethra	N.A.	pTisNxG3 squamous	0
9	Glansectomy + sLND	pT2N0G2 squamous	60	13	Excision	Outpatient (1)	Neoglans	4	PeINNxG3 squamous	0
10	Glansectomy + sLND	pT2N0G2 verrucous	62	82	Laser ablation	Outpatient (1)	Neoglans	15	pT1NxG1 squamous	0
11	Glansectomy	pT1NxG1 squamous	69	88	Laser ablation	Outpatient (1)	Neoglans	5	PeINNxG1 squamous	0
12	Partial glansectomy	pT1NXG3 squamous	70	45	Laser ablation	Outpatient (1)	Neoglans	9	PeINNxG3 squamous	0
13	Partial glansectomy + iLND	pT2N0G3 squamous	63	12	Excision	Inpatient (4)	Neoglans	18	pT1NxG2 squamous	2 (Antibiotic)

**Table 3 cancers-15-04807-t003:** Findings at follow-up of 13 penile cancer patients previously treated with glansectomy or partial penectomy, and subsequently treated with penile-sparing surgery for penile cancer recurrence between 2001 and 2022. iLND: inguinal lymph node dissection; sLND: sentinel lymph node dissection; NED: no evidence of disease; PeIN: penile intraepithelial neoplasia.

Patient	Type of Surgery	Histology	Follow-Up (Months)	Local Recurrence	Time Local Recurrence (Months)	Surgery Local Recurrence	Histology Local Recurrence	Regional Recurrence	Time Regional Recurrence (Months)	Surgery Regional Recurrence	Histology Regional Recurrence	Status
1	Circumcision + excision	N.A.	120	No	-	-	-	No	-	-	-	NED
2	Excision	pTaNxG1 squamous	44	Yes	7	Excision	pTaG1 squamous	No	7	sLND	pN0	NED
3	Circumcision + excision	N.A.	110	No	-	-	-	Yes	3	iLND	pN1	NED
4	Excision	pT1NxGx squamous	4	No	-	-	-	No	-	-	-	NED
5	Laser ablation	pT1NxGx squamous	71	No	-	-	-	No	-	-	-	NED
6	Laser ablation	N.A.	LOST	No	-	-	-	No	-	-	-	LOST
7	Excision	PeINNxG1 squamous	52	No	-	-	-	No	-	-	-	NED
8	Excision	pTisNxG3 squamous	29	No	-	-	-	No	-	-	-	NED
9	Excision	PeINNxG3 squamous	73	Yes	31	Penectomy	pT1G2 squamous	No	31	sLND	pN0	NED
10	Laser ablation	pT1NxG1 squamous	37	Yes	10	Penectomy	pT2G3 verrucous	No	-	-	-	NED
11	Laser ablation	PeINNxG1 squamous	9	No	-	-	-	No	-	-	-	NED
12	Laser ablation	PeINNxG3 squamous	0	No	-	-	-	No	-	-	-	NED
13	Excision	pT1NxG2 squamous	10	No	-	-	-	No	-	-	-	NED

## Data Availability

Not applicable.

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
