# Peer review of "Penile-Sparing Surgery for Tumour Recurrence after Previous Glansectomy/Partial Penectomy: Treatment Feasibility and Oncological Outcomes"

_cancers, 2023, doi:10.3390/cancers15194807_

Round 1

Reviewer 1 Report

In this retrospective study the authors present a single-center experience and oncological results in treating patients with penile cancer recurrence after glansectomy and partial penectomy. 

Penile reconstructive surgery is very challenging in penile cancer cases, and penile preserving procedures should be offered whenever feasible in order to obtain good functional, aesthetic and psychological outcomes, without compromising oncological outcomes. It is well known that these organ-sparing techniques are only indicated for small lesions where negative margins can be achieved. Otherwise, more radical approach is required. 

There are a few studies on penile cancer recurrence treatment. From this point of view, this is a valid study that provides new perspectives  in this challenging field. The report is very systematic, with detailed presentation of results (text and tables) and well prepared discussion. 

There are many obvious limitations of this study, and majority are defined in discussion section: retrospective design, lack of surgical margin status and lymphovascular invasion, patient quality of life with psychosexual and functional outcomes, and most important, comparison with other treatment group.

However, there are more issues that need to be addressed:

1. Median follow-up is 41 months, and median time after primary surgery to cancer recurrence was 56 months. A longer follow-up after PSS is necessary for objective oncological outcomes. 

2. Some illustrations would be helpful for readers. 

3. Selection criteria for patients with disease recurrence that are candidates for PPS is questionable, since there are no data on surgical margins in PSS. This is very important issue that needs attention and evaluation. 

4. Case number 8 is challenging and better insight is necessary for choice of treatment (excision) for urethra lesion. Location, size, no urethroplasty?

Overall, it seems that that strengths of the study are overweighted by its limitations. Based on these numerous limitations, the study and its results cannot support stated conclusions. Longer follow-up, defined selection criteria for PSS and comparison with other group is necessary to make such conclusions for safe oncological treatment. In this form, the study presents a single-center experience and conclusions should be re-defined accordingly.

Author Response

Reviewer #1

In this retrospective study the authors present a single-center experience and oncological results in treating patients with penile cancer recurrence after glansectomy and partial penectomy.

Penile reconstructive surgery is very challenging in penile cancer cases, and penile preserving procedures should be offered whenever feasible in order to obtain good functional, aesthetic and psychological outcomes, without compromising oncological outcomes. It is well known that these organ-sparing techniques are only indicated for small lesions where negative margins can be achieved. Otherwise, more radical approach is required.

There are a few studies on penile cancer recurrence treatment. From this point of view, this is a valid study that provides new perspectives in this challenging field. The report is very systematic, with detailed presentation of results (text and tables) and well prepared discussion.

There are many obvious limitations of this study, and majority are defined in discussion section: retrospective design, lack of surgical margin status and lymphovascular invasion, patient quality of life with psychosexual and functional outcomes, and most important, comparison with other treatment group.

However, there are more issues that need to be addressed:

Comment 1:
1. Median follow-up is 41 months, and median time after primary surgery to cancer recurrence was 56 months. A longer follow-up after PSS is necessary for objective oncological outcomes

Reply 1:

We thank the Reviewer for the pertinent comment. We agree with the Reviewer that longer follow-up is required for testing important oncological end points such as disease progression and cancer-specific mortality. In our preliminary report, we focused on treatment feasibility, setting of treatment and complications, rather than strong oncological outcomes. We agree with the Reviewer that our early oncological outcomes should be considered preliminary at best and that other series are required before implementing the systematic use of PSS for tumor recurrence after previous glansectomy/partial penectomy in daily practice. As suggested by the Reviewer, we added the following passage in the Discussion [page 8; lines 223-227]:”Last, the limited follow-up available for this patient cohort (median: 41 months) is not enough for testing major oncological endpoints such as tumor progression and cancer-specific mortality. In consequence, we advocate to test the oncological safety of another PSS procedure for tumor recurrence after previous glansectomy or partial penectomy in other series of patients with longer follow-up data”.

Comment 2:
2. Some illustrations would be helpful for readers.

Reply 2:

We thank the Reviewer for the pertinent comment. We added Figure 1 with two explicative cases.

Comment 3:
3. Selection criteria for patients with disease recurrence that are candidates for PPS is questionable, since there are no data on surgical margins in PSS. This is very important issue that needs attention and evaluation

Reply 3:

We thank the Reviewer for the pertinent comment. We agree with the Reviewer that accurate patient selection represent a key factor before recommending penile sparing surgery at disease recurrence after previous glansectomy or partial penectomy. Unfortunately, due to the rarity of the disease and, in consequence, due to the limited number of patients available for final analyses, only hypothetical considerations could be done. We modified the following passage in the Discussion by including the Reviewer’s suggestion (lack of information about surgical margins) [page 8; lines 180-195]: ”Due to the lack of specific recommendations for PSS after glansectomy/partial penectomy, accurate selection of candidates appears to be a key factor. Unfortunately, due to the lack of information about postoperative surgical margins and due to the small number of patients analysed, only hypothetical considerations could be derived from this analysis. Specifically, patient age, education, comorbidities, sexual life and compliance to strict follow-up schemes appear to be crucial. Moreover, other tumor characteristics, such as a long time to recurrence from previous surgery, small lesion size, low tumor T stage and grade, as well as recurrence location should be considered. Indeed, in our series, as compared to patients immediately treated with radical penectomy, patients treated with PSS had smaller and more superficial tumors. Moreover, time to disease recurrence was significantly lower for patient submitted to total amputation. Last, surgeon experience and hospital volume appear to be important when recommending PSS for recurrent disease. This said other reports testing the oncological safety and technical feasibility of PSS after glansectomy/partial penectomy are urgently required for optimizing patient selection and to promote wider use of PSS for this patient category.”

Comment 4:
4. Case number 8 is challenging and better insight is necessary for choice of treatment (excision) for urethra lesion. Location, size, no urethroplasty?

Reply 4:

We thank the Reviewer for the pertinent comment. Patient 8 had a small and superficial (Tis) lesion located in the distal urethra in the proximity of the neoglans. Due to patient preference and due to the lack of experience with urethroplasty at our centre, we performed a small excision of the lesion without compromising the function of the urethra. Patient was discharged with bladder catheter for 7 days after surgery. For sake of brevity, we did not add these information in the current version of the manuscript. We hope that the Reviewer will find our answer satisfactory.

Comment 5:
Overall, it seems that that strengths of the study are overweighted by its limitations. Based on these numerous limitations, the study and its results cannot support stated conclusions. Longer follow-up, defined selection criteria for PSS and comparison with other group is necessary to make such conclusions for safe oncological treatment. In this form, the study presents a single-center experience and conclusions should be re-defined accordingly.

Reply 5:

We thank the Reviewer for the pertinent comment. We already commented about follow-up time and selection criteria in, respectively, Comment 1 and Comment 3 of the same Reviewer. To accomplish with the Reviewer’s request, we plotted Kaplan-Meier plots depicting local and overall survival in patients with tumor recurrence after glansectomy/partial penectomy and treated with penile sparing surgery vs. radical penectomy (Supplementary Figure 1). Since the results observed in Supplementary Figure 1 could be a product of selection bias (i.e. lower overall recurrence rates in patients treated with penile sparing surgery), we decided to add this figure only as Supplemental material and not to extensively describe these findings in the manuscript.

We modified the text accordingly:

-[page 6; lines 147-150]:” Local and overall recurrence survival rates for patients previously treated with glan-sectomy/partial penectomy that underwent another PSS vs. radical penectomy for disease recurrence are depicted in Supplementary Figure 1.”

-[page 9; lines 242-248]:” . Last, we did not perform a systematic comparison between patients that were immediately treated with total penectomy at first local recurrence after glansectomy or partial penectomy vs. those re-treated with PSS. Specifically, we only reported Kalpan-Meier plots depicting local and overall recurrence survival rates in these two groups (Supplementary Figure 1), without extensively discuss our misleading findings (i.e. lower overall recurrence rates in PSS treated patients) that are, to our opinion, a product of selection bias.”

Last, as suggested by the Reviewer, we toned down our message in the Conclusions, that now read as follows [page 9; lines 250-253]:” We reported results of a preliminary analysis testing safety, feasibility and early oncological outcomes of PSS procedures for patients with local recurrence after previous glansectomy or partial penectomy. Stronger oncological outcomes should be tested in other series for optimizing patient selection.”

Reviewer 2 Report

The manuscript by Musi et.al. described a single center experience in penile sparing surgery for recurrent penile cancer following previous glansectomy and partial penectomy. They demonstrated that PSS was feasible in this clinical setting with satisfactory oncological outcomes and low rates of complications. Results from this study suggests that PSS can be a viable option for the management of recurrent penile cancer.

The authors performed survival analysis regarding local and overall recurrence following PSS and presented results in Figure 1. However there is no comparison group in this figure. Authors should consider including survival analysis of patients who were treated with total penectomy (n =22 as showed in supplementary table 1) so that readers can better appreciate the outcome of PSS in comparison to current standard of care.

Author Response

Reviewer #2

The manuscript by Musi et.al. described a single center experience in penile sparing surgery for recurrent penile cancer following previous glansectomy and partial penectomy. They demonstrated that PSS was feasible in this clinical setting with satisfactory oncological outcomes and low rates of complications. Results from this study suggests that PSS can be a viable option for the management of recurrent penile cancer.

Comment 1: 

The authors performed survival analysis regarding local and overall recurrence following PSS and presented results in Figure 1. However there is no comparison group in this figure. Authors should consider including survival analysis of patients who were treated with total penectomy (n =22 as showed in supplementary table 1) so that readers can better appreciate the outcome of PSS in comparison to current standard of care.

Reply 1:

We thank the Reviewer for the pertinent comment. As requested by the Reviewer, we plotted Kaplan-Meier plots depicting local and overall survival in patients with tumor recurrence after glansectomy/partial penectomy and treated with penile sparing surgery vs. radical penectomy (Supplementary Figure 1). Since the results observed in Supplementary Figure 1 could be a product of selection bias (i.e. lower overall recurrence rates in patients treated with penile sparing surgery), we decided to add this figure only as Supplemental material and not to extensively describe these findings in the manuscript. We hope that the Reviewer will find our answer satisfactory.

We modified the text accordingly:

-[page 6; lines 147-150]:” Local and overall recurrence survival rates for patients previously treated with glansectomy/partial penectomy that underwent another PSS vs. radical penectomy for disease recurrence are depicted in Supplementary Figure 1.”

-[page 9; lines 242-248]:” . Last, we did not perform a systematic comparison between patients that were immediately treated with total penectomy at first local recurrence after glansectomy or partial penectomy vs. those re-treated with PSS. Specifically, we only reported Kalpan-Meier plots depicting local and overall recurrence survival rates in these two groups (Supplementary Figure 1), without extensively discuss our misleading findings (i.e. lower overall recurrence rates in PSS treated patients) that are, to our opinion, a product of selection bias.”

Round 2

Reviewer 1 Report

Authors made necessary corrections according to comments and improved the paper.